# Case Report: A Case of Creutzfeldt–Jakob Heidenhain Variant Simulating PRES

**DOI:** 10.3390/diagnostics12071558

**Published:** 2022-06-27

**Authors:** Annibale Antonioni, Emanuela Maria Raho, Andrea Gozzi, Niccolò Cotta Ramusino, Edward Cesnik, Marina Padroni, Alessandro De Vito, Maura Pugliatti, Valeria Tugnoli

**Affiliations:** 1Unit of Clinical Neurology, Department of Neurosciences and Rehabilitation, University of Ferrara, 44121 Ferrara, Italy; emanuelamariaraho@gmail.com (E.M.R.); andrea01.gozzi@edu.unife.it (A.G.); niccol.cottaramusino@edu.unife.it (N.C.R.); pglmra@unife.it (M.P.); 2Neurology Unit, Department of Neurosciences and Rehabilitation, S. Anna University Hospital, 44124 Ferrara, Italy; csndrd@unife.it (E.C.); marina.padroni@ospfe.it (M.P.); a.devito@ospfe.it (A.D.V.); v.tugnoli@ospfe.it (V.T.)

**Keywords:** Creutzfeldt–Jakob disease (CJD), Heidenhain Variant, PRES, prion, neurodegenerative disease

## Abstract

The Heidenhain Variant of Creutzfeldt–Jakob disease (CJD) is an uncommon early clinical syndrome of the otherwise regular sporadic CJD, which belongs to the group of prion diseases caused by a transmissible agent, the misfolded form of the prion protein. The most characteristic symptoms of CJD are rapidly progressive cognitive impairment, typical motor manifestations and mental and behavioural changes. Conversely, in the Heidenhain Variant, different kinds of visual disturbances are observed at onset due to microvacuolar spongiform degeneration or, less frequently, confluent spongiform changes in the parieto-occipital area, detectable through brain MRI with hyperintensity in T2-FLAIR or DWI in the same areas. Since this an extremely rare condition with a heterogeneous clinical presentation, it may easily be misdiagnosed with other diseases at the earlier stages. Here, we describe the case of a patient initially diagnosed with posterior reversible encephalopathy syndrome (PRES), presenting with visual disturbances and headache at onset in a context of poorly controlled arterial hypertension. Subsequently, a rapid worsening of cognitive decline, associated with myoclonus and startle reaction led to further investigations, shifting the diagnosis toward a rapidly evolving neurodegenerative form. This hypothesis was also supported by EEG traces, MRI and CSF analysis. Finally, the clinical–instrumental evolution confirmed the diagnosis of Heidenhain Variant of CJD.

## 1. Introduction

Creutzfeldt–Jakob disease (CJD) is a neurodegenerative disorder belonging to the group of human prion diseases. There are four types of CJD: Sporadic (sCJD), Variant (vCJD), due to the ingestion of cattle affected by bovine spongiform encephalopathy (also called “mad cow disease”), Familiar or Genetic (gCJD) and Iatrogenic (iCJD), caused by the use of human grafts, surgical procedures or cadaver-derived therapies.

From a molecular point of view, there are different conformations of human PrPsc, called molecular strains, obtained using a proteinase K, which produces protein fragments of various sizes. In particular, these types can be subdivided according to the ratio of the bands generated by the protein digestion, which correspond to amino-terminally truncated degradation products generated from di-, mono-, or non-glycosylated PrPSc. Currently, four types of PrPSc are known based on molecular typing of the strain, but only one type is seen in variant CJD [1]. About sCJD, Parchi et al. proposed an initial classification based on the methionine/valine polymorphism at codon 129 of PRNP and the size of the prion protein resistant to protein digestion, which included four groups [2], which were later expanded to six phenotypes based on molecular–genetic features: MM1/MV1, VV1, MM2c, MM2t, MV2 and VV2 [3]. Further investigations have allowed correlation of these different types with diagnostic neuropathological features and the characteristic pattern of prion protein deposition [4].

The different types of CJD present various clinical characteristics, but the overall typical clinical picture consists of rapidly progressive encephalopathy with dementia, myoclonus and cerebellar ataxia [5]; sCJD is the most common form of CJD and is usually characterized by rapidly evolving dementia, with cerebellar ataxia, visual disturbances and myoclonus, leading to akinetic mutism. It also involves some atypical forms, such as an initial cerebellar form, the Brownell–Oppenheimer CJD, and the Heidenhain Variant, that occurs with visual impairment. Genetic forms of CJD involve clinico-pathological phenotypes with earlier onset and a more prolonged illness than sCJD. In these cases, genetic tests for family members should be considered. In the early stages, vCJD typically manifests with behavioural and psychiatric disorders in the absence of typical neurological features. During the course of the disease, involuntary chorea movements can occur [6].

Based on the above, CJD diagnosis is often challenging for the physician, considering that this disease can mimic other conditions. Definitive diagnosis can only be made through an autopsy of the patient’s brain, although there are some diagnostic tests that may help us to reach a possible or probable diagnosis [6]. For example, EEG shows sharp periodic tri-phasic or biphasic complexes on average in two-thirds of sCJD patients. Brain MRI can be useful, too. In particular, in T2-weighted, FLAIR and DWI sequences, hyperintensity can be observed in basal ganglia, thalamus or cortical areas in relation to different molecular types of CJD [6,7]. Finally, CSF analysis is often used in CJD diagnosis [7]. There are two principal biomarkers of neuronal damage to look out for: 14-3-3 protein and tau protein. However, it is important to bear in mind that these may be present as non-specific markers in several other clinical conditions, such as recent stroke, viral encephalitis, glioblastoma and paraneoplastic disorders among others. Furthermore, RT-QuIC, an emerging laboratory technique involving CSF samples, may provide a probable diagnosis of CJD as it allows detection of the pathological form of the prion protein [7].

Despite all the aforementioned diagnostic techniques, patient prognosis remains poor. Typically, progression is rapid up to death, which occurs on average after about 4-6 months following diagnosis. Treatment is, therefore, only palliative care [7].

In this paper, we present the case of a middle-aged man diagnosed with CJD Heidenhain Variant (HvCJD). This condition starts with visual disturbances of various kinds that lead to cortical blindness, which can be accompanied by myoclonus and dementia [8]. HvCJD, especially in the initial stage, can often be misdiagnosed with other clinical conditions, due to the non-specific clinical features and alterations on neuroimaging [9]. We describe a case of HvCJD initially misdiagnosed with PRES. Posterior reversible encephalopathy syndrome (PRES) is a clinical entity which presents many neurological symptoms, such as headache, visual disturbances, seizures, encephalopathy and focal neurological deficits. It is often associated with high blood pressure, which probably plays a prominent role in the pathogenesis of the disease, considering that the rapid development of hypertension compromises cerebral blood flow autoregulation, causing hyperperfusion, brain–blood barrier breakdown and vasogenic edema [10]. Posterior cerebral regions are more vulnerable to hyperperfusion because of the lack of sympathetic innervation to the vessels of the posterior circulation. A brain MRI typically shows focal regions of hyperintensity on T2-weighted sequences more frequently in the parietal and occipital lobes [11].

## 2. Case Presentation

A 66-year-old man from a closed community in northern Italy was admitted to our university hospital for visual disturbances and left-sided migraine headache without additional focal or lateral neurological deficits. These symptoms had started a few weeks before in association with high blood pressure. His medical history showed poorly controlled arterial hypertension, reduced coronary reserve and REM sleep behavioural disorders (RBD), which started about six months earlier and were treated with carbamazepine, subsequently suspended by the patient. He had a negative family history for neurodegenerative diseases. A brain CT did not show any recent lesions, while supra-aortic trunk echocolordoppler showed a 45% right internal carotid artery (ICA) stenosis, and a brain MRI documented signs of chronic microangiopathy. Owing to episodes of blurred vision, a campimetry examination was performed which showed a right lateral hemianopsia. He was then transferred to the neurology unit, where he underwent a brain contrast-enhanced MRI, the DWI sequences of which revealed a tenuous hyperlucency in the occipital cortical site, mainly parasagittal on the right (much less evident and shaded in the contralateral site), corresponding to a tenuous hyperintensity in FLAIR, slightly less recognizable if compared to the previous examination (Figure 1).

A likely evolution of posterior reversible encephalopathy syndrome (PRES) was hypothesized. During hospitalization, the patient complained of recurring headache, thus prompting rheumatologic evaluation and ultrasound of the temporal arteries, which excluded Horton’s arteritis. An EEG showed a non-specific finding characterized by recurrent paroxysmal abnormalities in the right posterior regions with diffusion to the homologous contralateral regions, at times with a periodic trend, which had not been observed in the tracing performed six months before during investigations carried out for RBD (REM sleep behaviour disorder) (Figure 2).

The patient was then discharged with a diagnosis of probable PRES, corroborated by the numerous episodes of increased blood pressure occurring during his stay in the hospital. Approximately one week later, as a result of continued visual impairment, the patient returned for further neurological examination; interestingly, other clinical findings were observed, namely, partial disorientation, marked latency in answering questions, dyscalculia, moderate deficit of mnestic functions, bilateral hypovisus with “blurred and fuzzy image”, slurred speech with aphasia and hints of constructive and gait apraxia. A further EEG revealed recurrent paroxysmal diphasic and triphasic abnormalities in the right posterior regions with diffusion to the contralateral homologous regions, sometimes in a periodic pattern (Figure 3), and the contrast-enhanced MRI was, on the whole, identical to the previous one.

Subsequently, clinical and instrumental observation showed a progressive worsening of the EEG tracing and neurological condition with perceptive errors, such as green vision for a few seconds, and hallucinations, speech deficits of moderate-severe degree, substantial paraphasias, use of circumlocutions and passe-partout words, myoclonus of the right upper limb and neck and dysmetria in the right half of the body. The presence of a rapidly evolving neurodegenerative disease was therefore conceivable. In the context of the differential diagnosis, the patient underwent investigations for autoimmune encephalitis, thyroid function test, anti-thyroperoxidase and paraneoplastic antibodies, with negative results. Furthermore, progressively documented EEG findings (i.e., periodic triphasic abnormalities) and the MRI pattern (bilateral hyperlucency in the occipital cortical site) also suggested prion disease, particularly in its occipital variant. For these reasons, the patient underwent a diagnostic rachicentesis with subsequent confirmation of a positive prion RT-QuIC. Furthermore, CSF analyses revealed values of 14-3-3 protein by ELISA equal to 29,100 AU/mL (interpreted positive, cut-off 23,400 AU/mL) and total tau by CLEIA (LUMIPULSE G600II) higher than 2000 pg/mL (reference range 90–450 pg/mL, value considered suspect for CJD > 1250 pg/mL). Analysis of the PRNP gene evidenced a genotype on codon 129 (RFLP or sequence) Methionine/Methionine (Met/Met). In the final stages, the patient presented monophasic speech, asynergy and prevalent right limb dysmetria during spontaneous motility and startle reaction. He therefore underwent palliative evaluation, and morphine therapy was initiated to relieve his suffering. In the end, intestinal subocclusion and multiple infectious events occurred, leading to the patient’s death as a result of septic shock. Finally, the autopsy on the patient’s brain confirmed the diagnosis of CJD (Creutzfeldt–Jakob Disease), Heidenhain Variant [9]. Of note, the brain autopsy revealed a prevalent MM1 phenotype characterized by microvacuolar spongiform changes, as well as the presence of the MM2c phenotype in the occipital lobe, identified by confluent spongiform changes (see Table 1 for a summary about the patient’s clinical history). 

## 3. Discussion

Creutzfeldt–Jakob disease is a transmissible neurodegenerative disease caused by the presence of a misfolded form of the prion protein which is normally present in cells and involved in numerous essential functions. Following this alteration, which may be sporadic, due either to genetic causes or the introduction of infected elements into the body (through feeding, surgical procedures, cadaver-derived therapies…), the misfolded prion protein induces a cascade of conversion from normal to pathological cellular isoforms, resulting in neurodegeneration and generation of further prion infectivity. Indeed, PrPSc is characterized by lower solubility and greater resistance to protein degradation due to its folded secondary structure and propensity to self-aggregate. The basic markers of the disease, common to all variants, are the presence of intra-neuronal vacuoles with a ‘spongy’ appearance, reactive gliosis and accumulation of plaques of abnormal proteins as well as co-factors responsible for stabilisation and infectivity. These alterations remain silent until clinical onset, which is typically characterized by a rapidly evolving dementia associated with cerebellar, psychiatric and behavioural disorders, as well as characteristic motor symptoms (myoclonias, startle reactions, etc.). There are some variants of this pathology, which are characterized by different symptoms at onset (e.g., Gerstmann–Sträussler–Scheinker, Kuru…); these include the Heidenhain Variant, which starts with visual disturbances of various kinds, such as vision loss and blurriness, followed by visual field defect, object distortion, loss of perspective and visual hallucinations; less frequently it presents with colour disturbance, visual agnosia, palinopsia and alexia/agraphia, due to alterations of the occipital and parietal cortex [8].

The diagnostic criteria are based on the patient’s clinical features (rapidly progressive cognitive decline, myoclonus, visual or cerebellar disturbances, pyramidal or extrapyramidal deficits and akinetic mutism) together with EEG investigations (typical periodic triphasic waves, predominantly in the posterior areas, usually evident in the later phases of disease), brain MRI (hyperintensity in T2-weighted images in parieto-occipital with less involvement of other cortical regions and the basal ganglia in DWI or FLAIR sequences) and the detection of 14-3-3 protein and Prion RT-Quic positivity on CSF. Depending on the presence of these findings, the diagnosis may be probable or possible, although the definite diagnosis of disease is only possible through autopsy of the brain [12]. Currently there is no treatment for these diseases, but clinical trials involving monoclonal antibodies that prevent the interaction between physiological and misfolded forms of the protein, as well as studies on the role of cofactors have been considered [13]. CJD enters into differential diagnosis with many pathologies, which can confuse the clinician, especially in the early stages of the disease. These include neurodegenerative, autoimmune/paraneoplastic, infectious, and toxic/metabolic disorders. In particular, according to a recent paper, the most common misdiagnoses are viral encephalitis, paraneoplastic disorder, depression, vertigo, Alzheimer’s disease, stroke, unspecified dementia, central nervous system vasculitis, peripheral neuropathy and Hashimoto encephalopathy [14,15,16]. In particular, the so-called rapidly progressive dementias (RPDs) should always be considered in the differential diagnosis of prion diseases. These include, for example, atypical manifestations of neurodegenerative diseases, but also some potentially treatable causes, such as encephalopathies of paraneoplastic origin, or infectious or neoplastic diseases [17].

Initially, in our case, the diagnosis of PRES was hypothesized for several reasons: the absence of a positive family history of hereditary neurodegenerative diseases, poorly controlled arterial hypertension despite polypharmacological therapy, associated with neuroimaging evidence of micro-angiopathy and several episodes of a marked rise in blood pressure during hospitalization; the first EEG pattern, although suggestive, was not specific in itself; the patient’s initial symptoms did not include signs of damage to cortical and cerebellar functions indicative of CJD [11].

There are two aspects to consider: firstly, the presence of RBD can sometimes be a marker of neurodegeneration, predictive for the development of synucleinopathies or, indeed, CJD [18]; the clinical evolution, neuroimaging data and EEG abnormalities seemed more consistent with the second hypothesis, so the diagnostic pathway for prion diseases was followed. Secondly, the first EEG (i.e., recurrent paroxysmal abnormalities in the posterior regions with diffusion to the homologous contralateral regions with a periodic trend), although not totally specific, could point towards the correct diagnostic hypothesis when the symptoms were characterised almost only by visual disturbances. Furthermore, the neuroradiological data, especially in this particular clinical context (i.e., poorly controlled arterial hypertension and onset of symptoms in the context of hypertensive crisis) and considering the large variability in neuroimaging of PRES, initially seemed consistent with this diagnosis, documenting a tenuous hyperlucency in the occipital cortical site in DWI sequences, corresponding to a tenuous hyperintensity in FLAIR. Indeed, although a brain MRI generally provides useful hints to distinguish PRES from the Heidenhain Variant, since the posterior subcortical involvement evidenced by the typical hyperintensity usually occurs during the course of PRES, and not in CJD, PRES can manifest itself with many less typical neuroradiological findings (e.g., restricted diffusion, lesions at different cortical sites, basal ganglia, etc.) [19]. Therefore, neuroimaging is generally not sufficient in the differential diagnosis of PRES, and also anamnestic and clinical data must be considered, which, in this particular case, proved to be misleading.

Finally, although the occurrence of PRES is becoming increasingly easier to identify, thanks to improved neuro-radiological investigations, its precise incidence is currently unknown, mostly because epidemiological studies tend to focus on patient populations at risk of developing it; in contrast, the Heidenhain Variant of CJD is an uncommon form of a very rare condition (only 4.9% in a sample of 370 people suffering from prion disease, where the incidence of CJD is 1-2/1,000,000 people) [9].

## 4. Conclusions

This case report emphasises the importance of considering the possibility of a prion disease (i.e., Heidenhain Variant of sCJD) in the differential diagnosis of visual disorders associated with suggestive anamnestic, clinical, neuroradiological and neurophysiological data. Such pathologies, although rare, are often misunderstood and misinterpreted, especially in the early stages, leading to diagnostic delays and inappropriate diagnostic, therapeutic and care pathways.

## Figures and Tables

**Figure 1 diagnostics-12-01558-f001:**
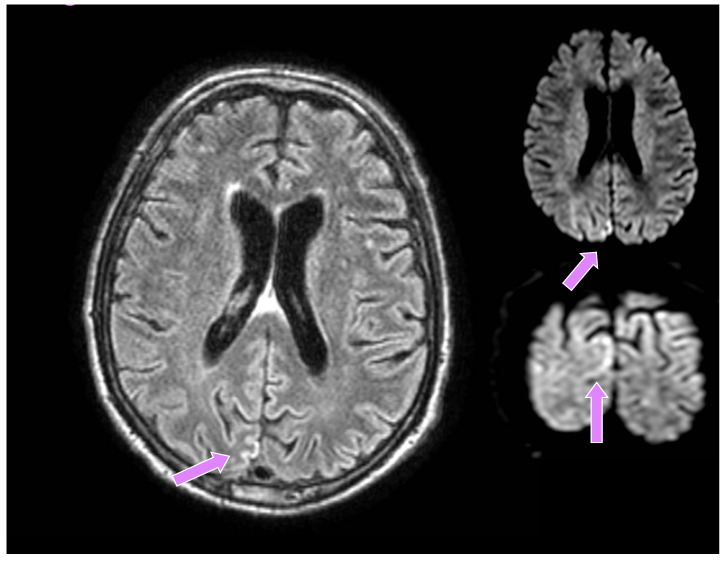
Brain MRI: altered signal in the occipital region, mainly parasagittal on the right, much less evident and shaded in the contralateral site (Seq. FLAIR, Axial and Coronal DWI), highlighted by pink arrows.

**Figure 2 diagnostics-12-01558-f002:**
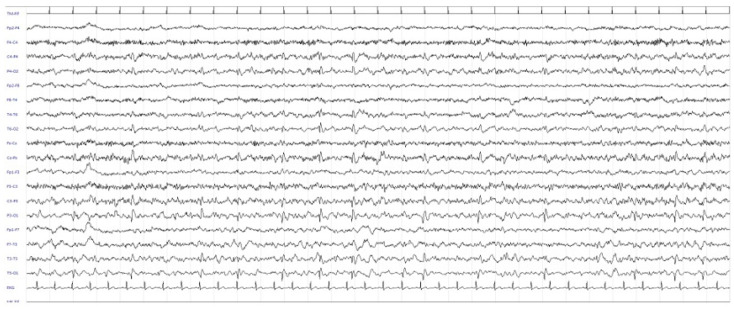
First EEG showing recurrent paroxysmal abnormalities in the right posterior regions with diffusion to the homologous contralateral regions with a periodic trend, which had not been observed in the tracing performed six months before during investigations carried out for RBD.

**Figure 3 diagnostics-12-01558-f003:**
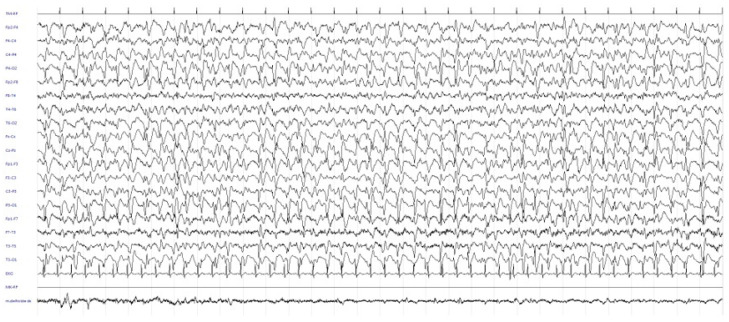
Recurrent paroxysmal diphasic and triphasic abnormalities in the right posterior regions with diffusion to the contralateral homologous regions, sometimes in a periodic pattern.

**Table 1 diagnostics-12-01558-t001:** Patient’s clinical history.

Date	Clinical Findings/Diagnostic Tests
Jan. 2021	REM behavioural sleep disorder, treated with carbamazepine.
May 2021	Visual disturbances and left migraine headache associated with high blood pressure. Brain CT scan and brain MRI were performed, only showing signs of chronic microangiopathy.
May 2021	Access to the neurology unit due to the presence of a right lateral hemianopsia. Brain contrast-enhanced MRI showed hyperintensity in the occipital cortex, mainly parasagittal on the right, in FLAIR sequences.
May 2021	Owing to the visual disturbances associated with headache and poorly controlled blood pressure, the patient was discharged with a diagnosis of PRES.
Jun. 2021	The patient returned for further neurological examination as a result of continued visual impairment and onset of other symptoms, such as disorientation and slurred speech with aphasia.
Jun. 2021	The EEG showed recurrent paroxysmal diphasic and triphasic abnormalities in the posterior regions. Contrast-enhanced brain MRI was, on the whole, identical to the previous.
Jul. 2021	Worsening of the clinical picture, with occurrence of perceptive errors, paraphasias, myoclonus of the right upper limb and neck and dysmetria in the right side of the body.
Jul. 2021	Diagnostic rachicentesis was performed and showed: confirmation of a positive prion RT-QuIC; 14-3-3 protein by ELISA equal to 29,100 AU/mL (interpreted positive, cut-off 23,400 AU/mL) Total tau by CLEIA (LUMIPULSE G600II) higher than 2000 pg/mL (reference range 90–450 pg/mL, value considered suspect for CJD > 1250 pg/mL). Analysis of the PRNP gene: genotype on codon 129 (RFLP or sequence) Methionine/Methionine (Met/Met)
Aug. 2021	Palliative therapy was started. The clinical course was complicated by various infective events, and, after a few weeks, the patient died as a result of septic shock
Nov. 2021	Autopsy on the patient’s brain confirmed the diagnosis of CJD Heidenhain Variant.

## Data Availability

Not applicable.

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
