# Peer review of "Case Report: A Case of Creutzfeldt–Jakob Heidenhain Variant Simulating PRES"

_diagnostics, 2022, doi:10.3390/diagnostics12071558_

Round 1
Reviewer 1 Report
Please find few comments and suggestions for this paper:
Introduction:
This section seems a bit lengthy, so a proper paragraphing and content minimization will make this paper more interesting.
Case Presentation
- In line 109, “left-sided migraine headache” would be appropriate
- How long is his RBD history?
- In line 116, full form of “ACI” should be mentioned.
- Was MRI not repeated after May 2021? .
- Repeat MRI and autopsy histopathology image could have been better
- Regarding differential diagnosis, please include autoimmune encephalitis panel, thyroid function test, TPO antibodies and paraneoplastic antibodies were done or not.
Discussion
- In line 186 and 198, please check for proper punctuation.
Conclusions
In line 225, comments on PRES could better be included in discussion portion.
Author Response
Dear Reviewer,
Thank you for your kind suggestions. We modified the manuscript according to your advice, in particular:
- We divided the introduction section into paragraphs and shortened it to avoid repetition with the discussion
- We modified line 109 according to your suggestion
- We added the duration of RBD history
- We spelt out the significance of ICA
- We added data about subsequent brain MRI in the text and table
- Unfortunately, we do not have the image from the autopsy investigation, but we have improved the neuroimaging data by including both FLAIR and DWI sequences and highlighting the sites of alteration
- We added a sentence regarding differential diagnosis, including autoimmune encephalitis panel, thyroid function test, TPO antibodies and paraneoplastic antibodies
- We corrected the punctuaction in the discussion
- We revised the discussion and the conclusion in order to highlight the strenght of our case report
Reviewer 2 Report
The authors present a case of sporadic Creutzfeldt-Jakob disease that was initially misdiagnosed as PRES. This case is an interesting example for the complexity of the clinical diagnosis of CJD and for pitfalls in the differential diagnosis of rapidly progressive encephalopathic syndromes. Its publication may be an important contribution and usefull for clinicians. However, I have some concerns regarding the article structure, the completeness of the discussion and some details of the presented background information.
Specific comments:
Lines 2-3: I am not a native speaker but I got the impression that the title itself is misleading. As far as I understood, the authors do not present CJD simulated by PRES but CJD simulating PRES.
Line 13: Calling the "Heidenhain Variant" a "disorder" that belongs to the group of prion diseases may also be misleading because it is usually not considered to be a distinct form or "variant" (although it is often named as such) of prion diseases. Instead, "Heidenhain variant" indicates one of the more uncommon early clinical syndromes of otherwise regular sporadic CJD.
Lines 31 and following: The introduction is very long and has the character of a mini review. Please check for redundancies within and with the discussion section.
Lines 42 and following: The authors describe the so-called "London" neuropathological classification in detail but do not discuss this classsification later in the context of Heidnehain variant or the particular case. Nor do they clarify differences/communalities with the Parchi classification. I would suggest to describe the Parchi classification first and then mention the alternative or "extended" London classification briefly.
Line 56: I wouldn't call the Brownell-Oppenheimer variant a purely cerebellar form of CJD because pathology and related symptoms spread to other regions during the disease course, just as in other CJD phenotpyes.
Lines 62, 66, 70: In my opinion, reference no. 5 (Mannix et al.) should not be considered as a reference for CJD criteria or specific symptoms. That article is a mini-review, not an expert consensus or original data. It might be considered when writing about decision making process for biopsy but the presented clinical diagnostic criteria are not original. Please consider:
Zerr et al. Updated clinical diagnostic criteria for sporadic Creutzfeldt-Jakob disease. Brain 2009; 132: 2659–68. (pre-RT-QuIC criteria, clinical syndrome, MRI and other diagnostic tools)
or
Vitali et al. Diffusion-weighted MRI hyperintensity patterns differentiate CJD from other rapid dementias. Neurology 2011; 76: 1711–9. (American MRI criteria)
or
Bizzi et al. Subtype Diagnosis of Sporadic Creutzfeldt-Jakob Disease with Diffusion Magnetic Resonance Imaging. Ann Neurol. 2021;89(3):560-572. doi:10.1002/ana.25983 (MRI criteria considering subtypes)
Line 76: RT-QuIC may be considered as "frequently used" but only from some point of view. It is increasingly used but still restricted to very few specialized centers and is not available in many countries.
Line 78: I discourage to use the term "mutated" in this context because it may rather indicate Prions from patients with PRNP mutations. In my opinion, just "...detection of the pathological Prion protein..." would be absolutely sufficient.
Lines 79-81: The controversy about 14-3-3 and t-tau (which one is better for the diagnosis of CJD?) is not new. There are several studies with contradictory results and recommendations. I think it is problematic to pick one study and to present its claims unreflected. On the other hand, an extended discussion of 14-3-3 vs. t-tau does not seem to be appropriate for the scope of this article. It may as well be omitted.
Lines 90-93: I do not agree that in HvCJD, other brain regions are always preserved. In fact, in MRI as well as in neuropathology, other regions are involved, too (e.g. Baiardi et al., ref. no. 9). Further, Hyperintensities may be seen on T2/FLAIR images but the main mechanism is restricted diffusion that can be detected with DWI/ADC and is usually also visually most impressive in DWI images.
Line 123: In this context, I recommend to show FLAIR AND DWI images in Figure 1.
Line 125, Figure 1 caption: The term "contrast-enhanced MRI" below this image may be misleading because it may give the impression that the image shows leasions with contrast agent enhancement. This would be something that clearly points to other diagnoses than CJD.
Lines 181 and following: Again, the discussion presents broad background information on several aspects of CJD, please check for unnecessary or redundant information. In my opinion, more information about the commonalities and differences of CJD and PRES, especially similarities and differences of MRI (!), and a clearer linking with the presented case data would be helpful.
Author Response
Dear Reviewer,
Thank you for your kind suggestions. We modified the manuscript according to your advice, in particular:
- We corrected the title
- We corrected the wording in relation to the Heidenhain Variant
- We divided the introduction section into paragraphs and shortened it to avoid repetition with the discussion
- We added a description of the Parchi classification
- We corrected the sentence about the Brownell-Oppenheimer variant
- We replaced reference n. 5 with one of those you kindly suggested
- We corrected the sentence on RT-QuIC
- In line 78, we replaced the word mutated with pathological
- We removed the sentence ragrding the controversy about 14-3-3 and t-tau according to your suggestion
- We pointed out that in Heidenhain's variant, although the main involvement is parieto-occipital, other brain areas can also be damaged
- We modified the figure 1 (DWI and FLAIR sequences) and its caption
- We revised the discussion and the conclusion in order to highlight the strenght of our case report
- We have tried to improve the text to make it more fluent and readable
Round 2
Reviewer 2 Report
I would like to thank the authors for considering my comments and I think that the revised manuscript has been substantially improved. Nonetheless, I still have some minor points:
Abstract
Line 13:
I wrote that the Heidenhein variant is one of the more uncommon clinical presentations of CJD. That was probably unclear or incorrect wording from my side. I think it would be misleading to describe it as “…one of the most uncommon…” clinical presentation of CJD. I would suggest to write “…is an uncommon early clinical…” instead.
Introduction
Lines 40-53:
In my opinion, this paragraph is still somewhat confusing. Again, this might be due to misleading wording in my initial comment and I wish apologize. Thus, I would like to present a detailed version of my suggestion for the paragraph to avoid further misunderstanding (moved or added text is bold):
From a molecular point of view, there are different conformations of human PrPsc, called molecular strains, obtained using a proteinase K, which produces protein fragments of various sizes. In particular, these types can be subdivided according to the ratio of the bands generated by the protein digestion, which correspond to amino-terminally truncated degradation products generated from di-, mono-, or non-glycosylated PrPSc. Currently, four types of PrPSc are kwon based on molecular typing of the strain but one type is only seen in variant CJD [Wadsworth et al. 2003].
About sCJD, Parchi et al. proposed a classification based on the methionine/valine polymorphism at codon 129 of PRNP and the size of the prion protein resistant to protein digestion, which included four groups [Parchi et al. 1996], which were later expanded to six phenotypes based on molecular-genetic features: MM1/MV1, VV1, MM2c, MM2t, MV2, VV2 [3]. Further investigations have allowed correlation of these different types with diagnostic neuropathological features and the characteristic pattern of prion protein deposition [4].
Lines 73-80: Please check the reference sequence in this section. I think the reference in line 74 (“…CJD diagnosis”) was probably meant be no. 7 (Hermann et al.) and not 8 (Muniz et al.). Similarly in line 80, the Muniz et al. reference is possibly misplaced.
Discussion
Line 202: “…with less involvement of the basal ganglia and white matter of the brain.” MRI white matter abnormalities are not a feature of CJD, at least on conventional DWI, T2 or FLAIR images. Thus, I suggest to write: “…with less involvement of other cortical regions and the basal ganglia.”
Line 221 and following: I think one sentence on common MRI differences between CJD and PRES would be helpful (e.g.: subcortical involvement of posterior signal hyperintensities is typically occurring during the course of PRES but in CJD).
Author Response
Dear Reviewer,
Thank you again for your valuable suggestions. We apologise for not having interpreted all your previous suggestions correctly. We have made the changes you suggested, in particular:
- In the abstract, we replaced the wording 'one of the most uncommon' with 'an uncommon';
- In the introduction, we changed the classifications according to your advice and corrected the references;
- In the discussion, we corrected the description of brain MRI and added a sentence explaining the typical difference in neuroimaging between CJD and PRES, but highlighting the variability of the latter and the consequent difficulties in differential diagnosis.